# Prevalence of low birth weight and associated factors in Ethiopia: An umbrella review of systematic review and meta-analyses

Neway Ejigu[1]*, Negussie Sarbecha[2], Kenbon Seyoum[1], Degefa Gomora[1], Girma Geta[1], Chala Kene[1], Sheleme Mengistu[1], Derese Eshetu[1], Yaregal Admasu[1], Telila Mesfin[3], Daniel Atlaw[4], Girma Beressa[4]

1 Department of Midwifery, School of Health Sciences, Madda Walabu University, Goba, Ethiopia,
2 Department of Biomedical Science, School of Medicine, Madda Walabu University, Goba, Ethiopia,
3 Department of Medicine, Collage Medicine and Health Sciences, Madda Walabu University, Goba, Ethiopia, 4 Department of Public Health, School of Health Sciences, Madda Walabu University, Goba, Ethiopia

* kookeetneway@gmail.com

## Abstract

Low birth weight (LBW) is one of the major causes of neonatal mortality and morbidity in low and middle-income countries (LMICs). Despite the goal of reducing newborn morbidity and mortality by 2030, low-income countries, including Ethiopia, still confront major challenges. Although various systematic reviews and meta-analyses (SRMA) have been conducted on LBW in Ethiopia, there is notable variation among their findings. This umbrella review aimed to consolidate inconsistent findings into a single summary estimate, providing a robust synthesis of evidence from systematic reviews and meta-analyses to bolster health policy development and planning in Ethiopia.Articles were retrieved on PubMed/Medline, Science Direct, Web of Science, HINARI, and Google Scholar. Assessments of Multiple Systematic Reviews checklist scores were used to assess the quality of the included SRMA studies. A random-effects model was used to estimate the overall effect size.A total of eleven SRMA studies (5 prevalence and 6 predictors) involving 190,492 neonates with an outcome of interest were included in the analysis. The summary estimate for the prevalence of LBW was 16% (95% CI: 13, 18%). Being prematurity [POR: 7.86; 95% CI: 5.79, 10.67], not attending antenatal care (ANC) [POR: 2.4, 95% CI: 1.49, 3.88], having pregnancy-induced hypertension (PIH) [POR: 4.2; 95% CI: 2.78, 6.36], being a rural resident [POR: 2.14, 95% CI: 1.56, 2.94], having a pregnancy interval < 24 months [POR: 2.96; 95% CI: 1.79, 4.9], not having iron-folic acid supplementation (IFAS) [POR: 0.38; 95% CI: 0.29, 0.5], and being a maternal age < 20 [POR: 2.02, 95% CI: 1.41, 2.9] were significantly associated with LBW. This umbrella review revealed more than three out of twenty neonates experienced LBW in Ethiopia. Being premature, not attending antenatal care, having pregnancy-induced hypertension, being

**Data availability statement:** All relevant data are within the article and its Supporting Information files.

**Funding:** The authors received no specific funding for this work.

**Competing interests:** The authors have declared that no competing interests exist.

a rural resident, having a pregnancy interval < 24 months, not having iron-folic acid supplementation and being a maternal age < 20 were significant predictors of LBW. Therefore, timely diagnosis, proper treatment, and follow-up of women at risk might combat the incidence of LBW in Ethiopia.

## Introduction

Low birth weight (LBW) is defined by the World Health Organization (WHO) as a birth weight of less than 2500 grams, regardless of gestational age [1]. LBW is a useful public health measure of poverty, quality of healthcare delivery, nutrition, and maternal health [2]. A newborn's prospects of long-term survival and development can be predicted by their birth weight [3]. Infants with LBW are more likely to experience growth retardation, developmental delay, infectious diseases, and non-communicable diseases (NCDs), which may occur during infancy, childhood, and later in life [4]. Furthermore, problems like hypothermia, hypoglycemia, prenatal asphyxia, respiratory distress, anemia, poor nutrition, infection, and hearing impairments are linked to LBW [5].

The healthcare systems, as well as affected families, bear substantial financial, social, and medical expenses due to LBW [6]. LBW is the leading cause of neonatal mortality and a predominant predictor of childhood morbidity and mortality [7–9]. Newborns with low birth weight are about 20 times more likely to die compared to normal birth weight [10]. Therefore, WHO has set a target of a 30% reduction in LBW by 2025 to protect the health of newborns and young children [11].

The etiology of LBW is the outcome of complex interactions of numerous environmental and physical factors [12]. Some of the factors that influence LBW are rural residence, extremes of maternal age, multiple pregnancies, obstetric complications, chronic maternal conditions, infections, and nutritional status [13–15]. In addition, inadequate antenatal care (ANC) follow-up, preterm birth (PTB), physically inactive, passive smoking, air pollution, female children, and low Iron intake may lead to higher rates of LBW[4,16–18].

In 2020 the prevalence of LBW is 14.7% worldwide, representing more than 19.8 million births a year [19]. Majority of LBW births occur in low and middle-income countries (LMICs) [20]. The pooled prevalence of LBW newborn babies' in Sub-Saharan Africa (SSA) was 13.9% [19] and in Ethiopia ranged from 10.06% [21] to 19.16% [22].

Reduction of neonatal mortality is one of the major Sustainable Developmental Goals (SDGs) for Ethiopia, aiming to lower the rate to below 12 per 1,000 live births by 2030 [23]. This goal supports SDG 3.2, which aims to eliminate preventable deaths among newborns and children under five. Despite significant efforts by the Ministry of Health (MoH) and non-governmental organizations (NGOs), LBW remains a major cause of neonatal morbidity and mortality in Ethiopia. This may be due to limited information available about the problem. Therefore, the availability of local information on the determinant of LBW has a major role in the management

and control of the case in the country. Moreover, this study will help the stakeholders and policymakers to reinforce the existing programs towards the problem. Even though, there are many systematic reviews and meta-analyses in Ethiopia, the findings are inconsistent. Therefore, the aim of this umbrella review was to consolidate inconsistent findings into a single summary estimate, providing a robust synthesis of evidence from systematic reviews and meta-analyses to inform health policy development and planning in Ethiopia.

## Methods

An umbrella review is a systematic review of systematic reviews, synthesizing only the highest quality evidence. This method offers a comprehensive resource, aiding policymakers, developing intervention strategies, establishing clinical guidelines, and evaluating healthcare evaluations [24,25]. This umbrella review was conducted following the methodology of an umbrella review of multiple systematic reviews, considered the most robust form of evidence [26,27]. The study was reported using the Preferred Reporting Items of Systematic Reviews and Meta-Analysis Protocols (PRISMA-P) checklist or guidelines [28] [S1 Checklist].

### Eligibility criteria

All eligible systematic reviews and meta-analyses (SRMA) using observational studies on LBW prevalence and its related factors were included. The pre-determined eligibility criteria were as follows: the population was newborn; exposure, predictors of LBW; study area, studies conducted in Ethiopia; study design, all SRMA studies; publication condition, both published and unpublished research; and language, studies reported in English. There were no restrictions on the publication dates of SRMA studies. We excluded narrative reviews, editorials, correspondence, abstracts, methodological studies, and literature reviews lacking a clear research topic, search strategy, or article selection criteria.

### Search strategies

Two authors (NE and NS) conducted a search for both published and unpublished SRMA from January to February 10/2024, for this umbrella review. For accessing relevant data about LBW, a comprehensive search was conducted through (PubMed/Medline, Science Direct, Web of Science, HINARI and Google scholar) databases. SRMA studies were identified through a comprehensive search using Boolean logic operators (AND, OR, NOT), Medical Subject Headings (MeSH), and relevant keywords in the aforementioned databases, based on PICOs questions. Key search terms were related to the ((("infant, low birth weight"[MeSH Terms] AND "Systematic Review"[Publication Type]) OR "Systematic Reviews as Topic"[MeSH Terms] OR "Systematic Review"[All Fields]) AND "Meta-Analysis"[Publication Type]) OR "Meta-Analysis as Topic"[MeSH Terms] OR "Meta-Analysis"[All Fields]) AND "Ethiopia"[MeSH Terms].

### Selection process

We exported all search results to the EndNote X8 citation system, where we removed duplicate articles to identify systematic reviews and meta-analyses that met the inclusion criteria. Two reviewers (NE and NS) independently screened the title and abstract against the predefined eligibility criteria. In the event of a disagreement, a consensus was reached to read the full lengths of the articles. The third reviewer (KS) was consulted when there was a discrepancy in order to make the final decision.

### Data extraction

Data from the included SRMA studies were extracted using a standardized data abstraction form created in excel spreadsheet. For each SRMA study, the following data were extracted: (a) identification data (first author's last name and publication year), (b) review aim (c) prevalence of LBW (d) risk factors for LBW (e) odds ratio (OR) along with 95% confidence intervals (CI) for the risk factors of LBW, (f) number of primary studies included within each SRMA study and their respective design type, (g) total number of sample size included, and (h) quality assessment methods.

### Missing data handling

We handled missing data by carefully considering the types of missing data and conducting sensitivity analyses.

### Quality assessment of the systematic review and meta-analyzed studies

All relevant systematic reviews and meta-analysis studies were assessed for quality using the AMSTAR-2 (Assessment of Multiple Systematic Reviews) tool, which comprises 16 items: nine noncritical and seven critical domains [29]. The critical domains include the protocol was registered before the review was started, the extent of the literature search, the justification for excluding particular studies, the risk of bias from the studies included in the review, the relevance of meta-analysis methods, taking into account the risk of bias when interpreting the review's findings, and the appraisal of the existence and likely consequences of publication bias [29]. The responses in the tool are listed as "yes," "partial Yes," "no," or "no meta-analysis conducted." For each of the included SRMA studies, two authors scored each of the 16 questions. The third reviewer resolved scoring disputes. An umbrella review quality based on AMSTRA-2 criteria was categorized as high, moderate, low, and critically low.

### Data analysis

The extracted data were exported to the statistical software R version 4.3.2 for analysis. The overall estimates of the prevalence of LBW and predictors were presented using forest plots utilizing the random effects model and the DerSimonian Liard method. OR along with a 95% CI were used to estimate the strength of the association between predictors and LBW. A narrative synthesis was used to present the findings of the included SRMA studies, followed by an overall meta-analysis. Heterogeneity test was assessed using the $I^2$ statistic tests of the included studies. The $I^2$ test statistics of 25%, 50%, and 75% were declared as low, moderate, and high heterogeneity, respectively [30]. Publication bias could not be assessed due to the inclusion of only five studies.

## Results

### Search findings

The database search yielded a total of 918 articles [S1 File]. Of these, 322 articles from the identified studies were removed due to duplication. Subsequently, 578 out of 596 articles were excluded after reviewing the title and abstract. Upon a full-text review of the remaining 18 articles, seven SRMA studies were excluded for various reasons: four studies [31–34] did not consider the required outcome, two studies [35,36] were conducted on high risk populations, and one study [37] did not meet the inclusion criteria [S1 Table]. Finally, a total of 11 SRMA studies [21,22,38–46] were included in the current umbrella review [Fig 1].

### Characteristics of included systematic review and meta-analysis studies

All systematic review and meta-analysis (SRMA) in this umbrella review were based on 259 primary observational studies [21,22,38–46]. Among these, there were 148 cross-sectional studies, 82 case–control studies, and 29 cohort studies. In these umbrella review, the median number of studies included in each SRMA with outcomes of interest was 24 studies, ranging from 5 studies [41] to 43 studies [22]. The median number of participants in these SRMA with outcomes of interest was 10,989, ranging from 2,526 [41] to 55,085 [40]. Across the 11 SRMA studies, a total of 217,722 neonates were included, with 190,492 neonates had the outcome of interest LBW. Regarding the publication of the included SRMA one research was unpublished [22], while the remaining ten studies were published within the last five years. Out of the included SRMA studies, four investigated both the prevalence and determinants of LBW [22,40,42,43], one solely reported the prevalence [21] and six only reported factors associated with LBW in Ethiopia [38,39,41,44–46] [S2 Table]. As per the SRMA studies included, the reported prevalence estimate of LBW in Ethiopia varied from 10.06% (95% CI: 7.2, 12.91%)

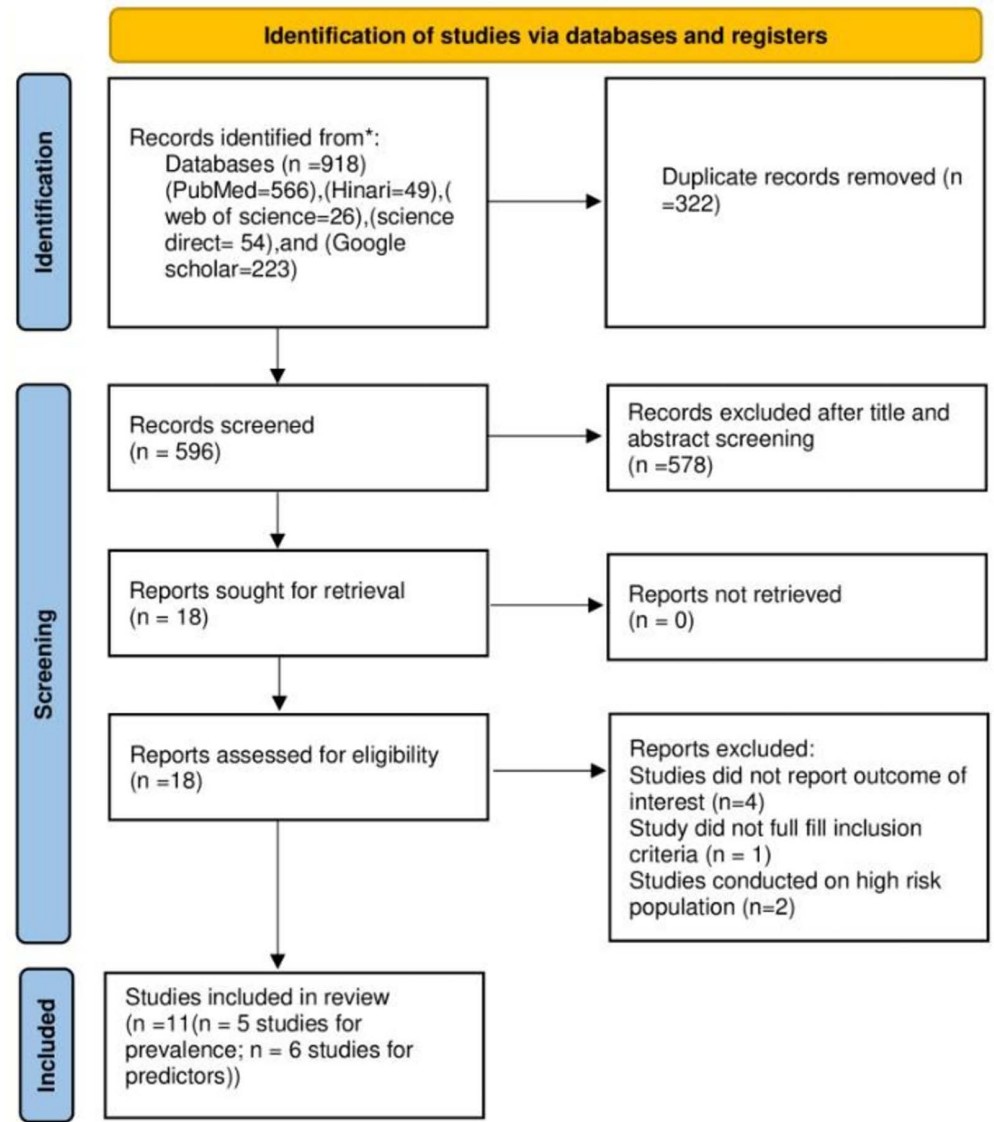

**Fig 1. PRISMA Flow diagram for searching, screening and identification of SRMA studies.**

[21] to 19.16% (95% CI: 18.01, 65.99%) [22]. The earliest article included in this umbrella review was published in 2018 [40] and the most recent was published in 2023 [39]. The methodological quality of the included SRMA studies was evaluated using the AMSTAR-2 critical appraisal checklist, with five articles rated as high quality, four as moderate and two categorized as low quality [S3 Table].

## Primary studies

Primary studies are the original research studies providing firsthand data, distinct from systematic reviews or meta-analyses that synthesize findings from multiple primary sources. To determine whether the reviews were based on the same primary evidence, primary studies within the eleven SRMA studies that were included were mapped. A total of 259 primary studies were included in the review. Only 226 of the 259 main papers that were included in the 11 SRMA studies

**Table 1.  Characteristics of included and excluded systematic review and meta-analysis studies.**

| Table 1 a: Characteristics of included systematic review and meta-analysis studies | | | | | | | | | | | |
|---|---|---|---|---|---|---|---|---|---|---|---|
| Authors & year of publication | Publication period of primary studies | Included number of primary studies and design | Sample size | Reported prevalence | Primary outcome of the review | Reported factors | Quality assessment | Name of data extractor | Date of data extraction | Quality score | URL |
| Habtegiorgis et al., [22] | 2018-2023 | 43 studies (25 cross-sectional,5 cohort and 13 case control) | 19,889 | 19.16% (95% CI: 16.25, 22.07) | LBW | Maternal age<20 (OR=2.46, 95% CI: 1.96, 3.09), ANC follow up (OR=3.00, 95% CI: 1.86, 4.84), GA<37 weeks (OR=9.68, 95% CI: 5.88, 15.94), birth interval<24 months (OR=3.97, 95% CI: 1.13, 13.97), PIH (OR=2.83, 95% CI: 1.34, 5.97), and maternal anemic status (OR: 3.32, 95% CI: 1.14, 9.69) | NOS | NE &NS | 23/2/2024 | Medium | https://papers.ssrn.com/sol3/papers.cfm?abstract_id=4417075 |
| Endalamaw et al., 2018 [40] | 1989-2017 | 33 studies (22 cross-sectional, 8 cohort and3 case control) | 55,085 | 17.3% (95% CI: 14.1–20.4) | LBW | Maternal age<20 years (AOR=1.7, 95% CI:1.5–2.0), pregnancy interval<24 months (AOR=2.8, 95% CI: 1.4,4.2), BMI<18.5kg/m$^2$ (AOR=5.6, 95% CI: 1.7,9.4), and GA<37 weeks (AOR=6.4, 95% CI: 2.5,10.3) | JBI | NE &NS | 23/2/2024 | High | |
| Gedefaw et al., 2020 [21] | 2013-2019 | 17 studies (14 cross-sectional and 3 case control) or 13 studies(cross-sectional) | 8,846 | 10.06% (95% CI; 7.21–12.91) | LBW | None | JBI | NE &NS | 23/2/2024 | High | |
| Katiso et al., 2020 [42] | 1990-2017 | 28 studies (17 crossectional, 8 cohort,and 3 case control) | 50,110 | 14.1% (95% CI: 11.2, 17.1) | LBW | Female babies (OR=1.5, 95% CI: 1.2, 1.7), GA<37 weeks (OR, 4.7, 95% CI: 1.5, 14.5), not attending ANC (OR,1.7 (95% CI:1.4, 2.2), PIH (OR=6.7, 95% CI:3.5, 12.9), and rural areas (OR=1.8, 95% CI:1.2, 2.6) | JBI | NE &NS | 24/2/2024 | Medium | |
| Tamirat et al., 2020 [43] | 2000-2018 | 16studies(9,cross-sectional, 4 case control, and 3 cohort) | 20,484 | 18% (95% CI: 13.9%, 22.2%) | LBW | GA<37weeks (AOR=7.8, 95% CI: 4.7, 12.95), no ANC (AOR=3.39, 95% CI: 1.65, 6.98), rural residence (AOR=2.44, 95% CI: 1.94,3.08) and women with medical illness during pregnancy (AOR,4.36; 95% CI: 2.55, 7.44) | JBI | NE &NS | 24/2/2024 | Medium | |
| Getaneh et al., 2020 [44] | 2005-2020 and one unpublished study | 25 studies (15 cross-sectional,5 cohort,and 5 case control) | 4,279 | NA | LBW | PIH (OR=3.89, 95% CI: 2.66, 5.69) | JBI | NE &NS | 24/2/2024 | High | |

*(Continued)*

**Table 1.** (Continued)

| Table 1 a: Characteristics of included systematic review and meta-analysis studies | | | | | | | | | | | |
|---|---|---|---|---|---|---|---|---|---|---|---|
| Zenebe et al., 2021 [45] | 2015-2020 | 24 studies (9 cross-sectional and 15 case control) | 10,967 | NA | LBW | IFAS (OR = 0.37, 95% CI: 0.25, 0.55) | – | NE &NS | 24/2/2024 | Low | |
| Tegegne et al., 2021 [46] | 2014-2020 | 24 studies (9 cross-sectional and 15 case control) | 10,989 | NA | LBW | IFAS (OR = 0.39, 95% CI: 0.27, 0.59) | – | NE &NS | 20/2/2024 | Low | |
| Alebachew et al., 2021 [38] | 2015-2020 | 7 studies (4 case control and 3 crossectional) | 2506 | NA | LBW | Alcohol use (AOR = 9.39, 95% CI: 2.84, 15.94), Khat users (AOR = 3.19, 95% CI: 1.01, 5.37), antenatal cigarette smokers (AOR = 4.36, 95% CI: 1.75, 6.98), narghile users (AOR = 20.1, 95% CI: 3.94, 103) | NOS | NE &NS | 20/2/2024 | High | |
| Gebrahana et al., 2022 [41] | 2014-2022 | 5 studies (2 case control and 3 crossectional) | 2526 | NA | LBW | Intimate partner violence (AOR = 3.69, 95% CI: 1.61, 8.50) | NOS | NE &NS | 20/2/2024 | Medium | |
| Demelash et al., 2023 [39] | 2015-2021 | 8 studies (5 cross-sectional and 3 case control) | 13,352 | NA | LBW | Prenatal biomass fuel use (OR = 2.10, 95% CI: 1.33, 3.31), no separate kitchen (OR = 2.48, 95% CI: 1.25, 4.92), Active cigarette smoker women (OR = 4.11, 95% CI: 2.82,5.89), passive smoker women (OR = 2.63, 95% CI: 1.09, 6.35) | NOS | NE &NS | 20/2/2024 | High | |
| Table 1 b: Characteristics of excluded systematic review and meta-analysis studies | | | | | | | | | | | |
| Authors & year of publication | Reason for exclusion | | | | | | | | | | |
| Leta et al.,2022 [31] | Did not consider the required outcome of interest | | | | | | | | | | |
| Techane et al.,2022 [32] | Did not consider the required outcome of interest | | | | | | | | | | |
| Shiferaw K et al.,2021 [33] | Did not consider the required outcome of interest | | | | | | | | | | |
| Teshome A et al.,2016 [34] | Did not explain clearly the required outcome of interest | | | | | | | | | | |
| Mersha et al., 2019 [35] | Conducted among high risk populations (i.e., among hypertensive patient) | | | | | | | | | | |
| Bayih WA 2021., [36] | Conducted on high risk populations(i.e., among antenatal substance user) | | | | | | | | | | |
| Bililign et al.,2018 [37] | Did not meet the inclusion criteria(i.e., narrative study) | | | | | | | | | | |

ANC: Antenatal care; AOR: adjusted odds ratio; BMI: body mass index; CI: confidence interval; GA: gestational age; IFAS: iron with folic acid supplementation; JBI: Joanna Briggs Institute; LBW: low birth weight; NOS: Newcastle-Ottawa Scale; PIH: pregnancy-induced hypertension

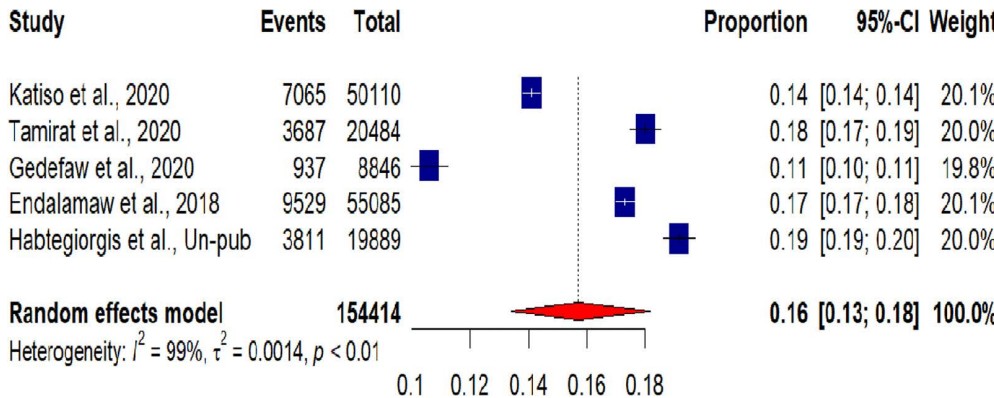

**Fig 2. An umbrella review of systematic review and meta-analysis studies on the prevalence of low birth weight in Ethiopia, 2024.**

reported LBW. We identified seventy-nine different primary publications after critically evaluating the eleven SRMA studies that were included. This suggests that at least two SRMA studies shared primary studies. For instance, six SRMA studies [38–40,42,43,46] included one primary study [47], two SRMA studies [40,42] included twenty two primary studies, and five SRMA studies [40,42,43,45,46] included one primary studies [48]. Any umbrella review should always have some overlap, which is one of the shortcomings of this study.

In contrast, thirty three primary studies were specific to Habtegiorgis et al. [22], thirteen to Getaneh et al. [44], nine to Gedefa et al. [21], five each to Katiso et al. [42] and Endalamew et al. [40], four each to Gebrahana et al. [41] and Tamirat et al. [43], three to Demelash et al. [39], two to Alebachew et al. [38] and one to Zenebe et al. [45] indicating that there was no overlapping of data from the aforesaid seventy-nine primary studies resulting in the different prevalence of LBW among the included eleven SRMA studies, which in turn necessitated the conduct of this umbrella review **[S4 Table]**.

## Meta-analysis of the prevalence of low birth weight

Out of the 11 SRMA studies, five [21,22,40,42,43] reported the prevalence of low birth weight and were included in the meta-analysis. The overall pooled prevalence of low birth weight as defined by an infant having a birth weight of less than 2500 gram in Ethiopia was 16% (95% CI 13, 18; $I^2$ = 99%) based on the umbrella review of these studies **[Fig 2]**.

## Meta-analysis on the association between prematurity and low birth weight

Among the included SRMA studies, ten [22,38–46] examined several factors associated with LBW. Four studies [22,40,42,43] examined the association between prematurity (delivered before 37 weeks of gestation) and LBW in Ethiopia. Research by Katiso et al. [42] revealed that preterm babies had a 4.7 fold more likely to be LBW compared to term babies [OR = 4.7 (95% CI 1.5, 14.5]. Studies by Tamirat et al. [43], Endalamaw et al. [40] and Habtegiorgis et al. [22] also showed that there was a statistical association between prematurity and LBW [OR = 7.8, 95% CI 4.7, 12.95], [OR: 6.4; 95% CI 2.5, 10.3], and [OR: 9.68; 95% CI 5.88, 15.94], respectively. The overall pooled estimate indicated that preterm babies were 7.86 times more likely to be LBW compared to their counter parts [POR: 7.86; 95% CI 5.79, 10.67].

## Meta-analysis on the association between antenatal care and low birth weight

Women who did not attend ANC were significantly associated with LBW in three SRMA studies [22,42,43]. According to the SRMA study conducted by Katiso et al. [42] women who did not attend ANC were 1.7 times more likely to have LBW babies as compared to women who did receive ANC [OR = 1.7, 95% CI 1.4, 2.2]. Other SRMA studies carried out by

Tamirat et al. [43] and Habtegiorgis et al. [22] also showed that there was a statistical association between ANC attendance and LBW [OR = 3.39, 95% CI 1.65, 6.98], and [OR: 3.0; 95% CI 1.86, 4.84], respectively. The overall effect estimates revealed that the odds of having LBW were 2.4 times higher among women who did not attend ANC compared to those who did receive ANC [POR = 2.4, 95% CI 1.49, 3.88].

## Meta-analysis on the association between pregnancy induced hypertension and low birth weight

Pregnancy induced hypertension (PIH) was significantly associated with LBW in three SRMA studies [22,42,44]. A study by Katiso et al. [42] indicated that mothers with PIH were 6.7 times more likely to have LBW babies [OR = 6.7 (95% CI 3.5, 12.9]. Research by Habtegiorgis et al. [22] and Getaneh et al. [43] also demonstrated a significant association between PIH and LBW [OR = 2.83, 95% CI 1.34, 5.97] and [OR: 3.89; 95% CI 2.66, 5.69], respectively. The overall pooled estimate revealed that the odds of LBW in women with diagnosed PIH were 4.20 times higher compared to normotensive women [POR: 4.2; 95% CI 2.78, 6.36].

## Meta-analysis on the association between residence and low birth weight

Maternal place of residence was associated to LBW in two SRMA studies [42,43]. A study by Katiso et al. [42] revealed that mothers in rural areas were 1.75 times more likely to have LBW babies [OR = 1.75 (95% CI 1.19, 2.56]. Similarly, a study by Tamirat et al. [43] indicated that there was a statistical association between maternal residence and LBW [OR = 2.44, 95% CI 1.94, 3.08]. The overall effect estimates suggested that the odds of LBW were 2.14 times higher among women residing in rural areas compared to their counterparts [POR = 2.14, 95% CI 1.56, 2.94].

## Meta-analysis on the association between pregnancy interval and low birth weight

Women who give birth in less than 24 month interval are significantly associated with LBW in two SRMA studies [22,40]. Research by Endalamaw et al. [40] revealed that the odds of infants born within less than a 24-month birth interval were nearly three times to have LBW [OR = 2.8 (95% CI 1.4, 4.2]. Similarly, a study by Habtegiorgis et al. [22] also showed a statistical association between birth in less than 24 month interval and LBW [OR = 3.97, 95% CI 1.13, 13.97]. The overall pooled estimate suggests that women who give birth with in less than a 24-month interval are 2.96 times more likely to have LBW baby [POR = 2.96 (95% CI 1.79, 4.9] compared to woman who give birth at greater than a 24-month birth interval.

## Meta-analysis on the association between iron/folic acid supplementation and low birth weight

Two SRMA studies [45,46] were included in the meta-analysis of the impact of iron/folic acid supplementation (IFAS) on LBW in Ethiopia. Study by Zenebe et al. [45] revealed that mothers who received IFAS had 0.61 lower odds of delivering LBW babies [OR = 0.39 (95% CI 0.27, 0.59]. Similarly, a study by Tegegne et al. [46] revealed a significant association between IFAS and LBW [OR = 0.37, 95% CI 0.25, 0.55]. The overall pooled estimate showed a 62% decrease in LBW odds among IFAS recipients [POR = 0.38, 95% CI 0.29, 0.5] compared to those do not receiving IFAS.

## Meta-analysis on the association between maternal age and low birth weight

Women who gave birth before the age of 20 had a significant association with LBW in two SRMA studies [22,40]. A study by Endalamaw et al. [40] revealed that woman who gave birth before the age of 20 were 1.7 times more likely to have LBW compared to those gave birth after the age of 20 [OR = 1.7, 95% CI 1.5, 2.0]. Similarly, a study by Habtegiorgis et al. [22] indicated a statistical association between women who gave birth before the age of 20 and LBW [OR = 2.46, 95% CI 1.96, 3.09]. The overall effect estimates revealed that the odds of LBW were 2.02 times higher among women who gave birth before the age of 20 compared to their counterparts [POR = 2.02, 95% CI 1.41, 2.9] [Fig 3].

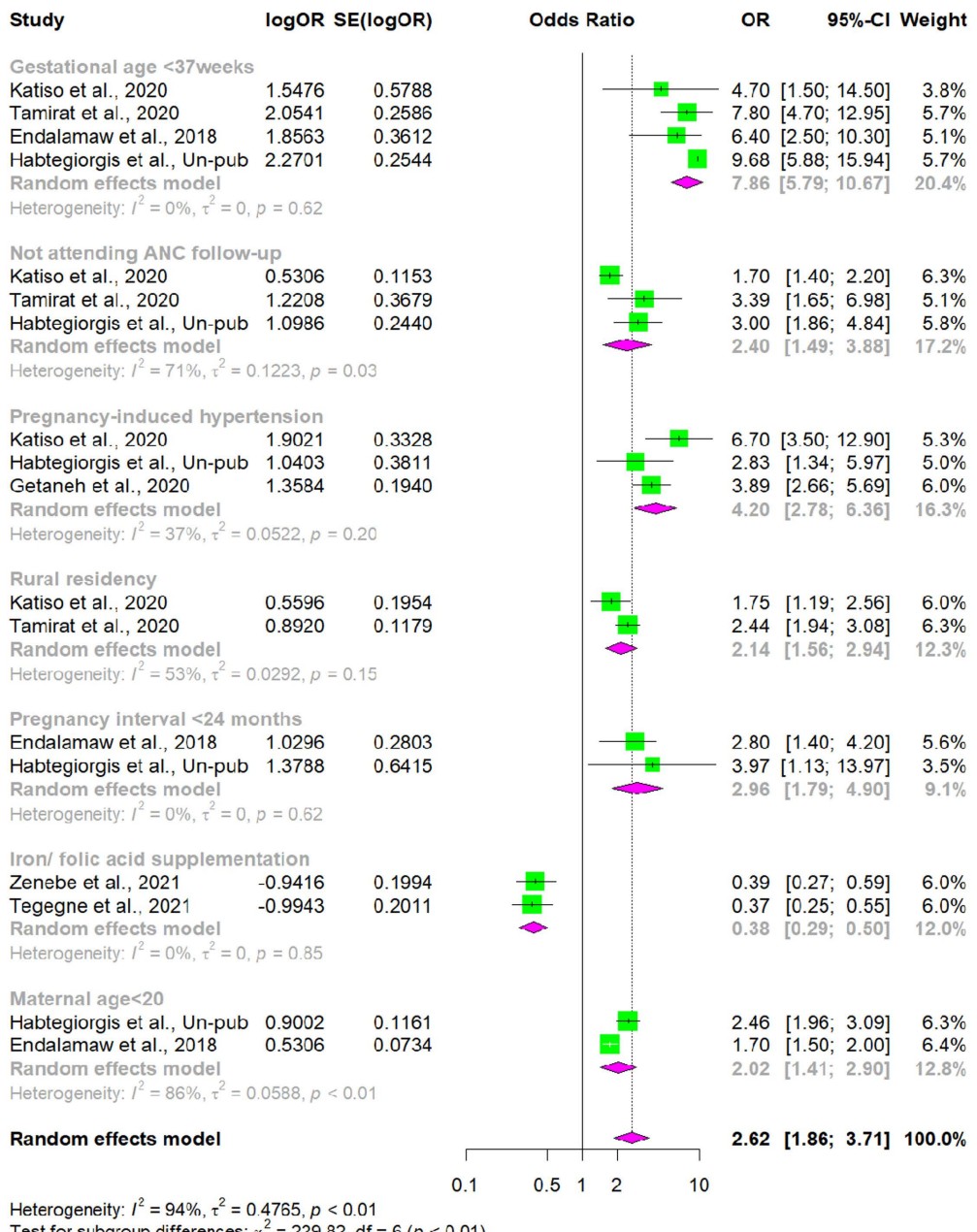

**Fig 3. An umbrella review about the pooled effects of risk factors on low birth weight in Ethiopia, 2024.**

## Discussion

Eleven SRMA studies investigated LBW in Ethiopia. SRMA studies are crucial to provide substantial evidence for decision-making in health programs and efforts. Yet, as the number of individual reviews increases, it may become exhausting for the information user. Therefore, this umbrella review aimed to provide more consistent conclusions by summarizing the eleven SRMA studies on LBW into a single document. The findings indicated that LBW was widely prevalent

and a serious public health concern in Ethiopia. Furthermore, pregnancy-induced hypertension, antenatal care, maternal age, prematurity, pregnancy interval, and place of residency were found to be statistically significant in predictors of LBW.

The five SRMA in this umbrella review [21,22,40,42,43] revealed that the overall pooled prevalence of LBW in Ethiopia was 16% (95% CI 13, 18%). This study finding agreed with studies from Africa [49] and global systematic analyses [13], but the result were higher than those conducted in SSA countries [50], and Iran [51]. The possible explanation for the variations may be the difference in sample size and differences in geographical variation, which might have caused differences in health service coverage and utilization.

We found a significant association between LBW and PTB. Infants who were delivered before 37 weeks of gestation were 7.86 times more likely to have a LBW compared to those delivered after 37 weeks of gestation. This might be due to the fact that several fetal organs typically mature by the end of 37 weeks of gestation, so infants born before this period have less time in utero to gain weight. This finding was supported by study in Indonesia [52].

Antenatal care follow-up was significantly associated with LBW. The odds of delivering babies with LBW among women who did not attend ANC follow-up were 2.4 times higher than those who did attend ANC. The reason behind this could be that ANC follow-up allows for the monitoring of fetal well-being and prompt intervention in case of feto-maternal issues being detected and managed or referred at an earlier stage. Moreover, ANC offers services such as regular nutritional and medical guidance, along with the provision of iron supplements for the health of both the mother and the fetus. This finding is consistent with studies from Africa [49], Nepal [53], Malaysia [54], and Asia [55]. Therefore, particular emphasis should be given to scaling up regular antenatal care follow up, health education, early detection, and intervention of obstetric complications with the help of Community Health Workers/ Volunteers (CHW/Vs).

In this study, women with PIH had 4.20 times higher odds of delivering babies with LBW compared to normotensive women. This might be due to endothelial cell injury and vasoconstriction of blood vessels, resulting in reduced utero-placental blood perfusion leading to LBW [56]. This finding is in line with the WHO secondary analysis survey conducted in LMICs [57], Indonesia [52], and Malaysia [54]. Therefore, the provision of timely and effective care for women experiencing these complications is crucial.

We observed that the odds of delivering babies with LBW among women residing in rural areas were 2.14 times higher compared to their counterparts. The possible reason might be that the lack of access to health care services for women in rural areas, leading to a lack of awareness regarding pregnancy, childbirth, and associated risks. Additionally, cultural practices in rural areas greatly impact women's nutritional status by the prohibiting of essential foods and drinks. This finding is agreed with studies from Jordan [58], Indonesia [52], and Malaysia [54].

In this research, pregnancy interval was significantly associated with LBW. Women who gave birth before 24 months had 2.96 times higher odds of delivering babies with LBW compared to those with birth interval greater than 24 months. The possible explanation might be due to maternal depletion syndrome or pregnancy-breastfeeding overlaps that deplete maternal resources via breastfeeding for the child already born and trans-placental sharing for the fetus in the womb. This, in turn, reduces the nutritional requirements of the fetus in the womb and subsequently results in LBW. This finding is consistent with studies conducted in Jordan [58] and Indonesia [52]. Therefore, it is crucial to prioritize appropriate ANC visits and postnatal care observations.

We observed a significant association between LBW and IFAS. The odds of LBW among women who received IFAS decreased by 62% compared to those who did not receive IFAS. The exact physiological mechanism through which iron supplements affect birth weight remains unclear; however, two hypotheses exist. First, oxidative stress on fetal growth is generated by changes in norepinephrine, cortisol, and corticotrophin caused by iron deficiency anemia, which iron supplements can alleviate. Second, iron supplementation increases appetite, which improving maternal nutritional status and consequently contributes to an increasing infant birth weight [53]. This finding is consistent with study in Malaysia [54]. Therefore, increasing utilization of antenatal care and iron supplementation during pregnancy shall be strengthened.

In this study, maternal age was significantly associated with LBW. Women who gave birth before the age of 20 were 2.02 times more likely to deliver babies with LBW compared to women who gave birth after the age of 20. This age group may represent teenage pregnancies that are more prone to pregnancy-related high blood pressure and anemia, leading to preterm labor and delivery. Furthermore, pregnancy at this age might leads to less attention to pregnancy-related problems, nutritional intake, and utilization of health care services, often due to unplanned and/or unwanted pregnancy, which frequently result in LBW infants. This finding is comparable with a multicounty study conducted by WHO in 29 countries [57], Malaysia [54], and Indonesia [52].

## Limitation of the review

This umbrella review may be constrained by the overlap of the primary studies with those considered by the systematic review and meta-analysis. Additionally, a limitation of this umbrella review is the reliability of the included unpublished and non-peer-reviewed publications. Further, the data should be reported with caution, because of the high heterogeneity.

## Conclusion

This umbrella review revealed more than three out of twenty neonates experienced LBW in Ethiopia. Being prematurity, not attending ANC, having PIH, being a rural resident, having a pregnancy interval < 24 months, not having IFAS, and being a maternal age < 20 were significant predictors of LBW. Therefore, the MoH should tackle the factors contributing to LBW by effectively guiding and enforcing obstetric care providers and health extension workers to deliver comprehensive community education on the impacts of short birth intervals, ANC follow-up, and IFAS. Moreover, early identification and management of high-risk pregnancies, such as PIH, are essential to mitigate the prevalence of LBW.

## Supporting information

**S1 Checklist. Preferred Reporting Items of Systematic Reviews and Meta-Analysis Protocols (PRISMA-P) checklist or guidelines.**
(DOC)

**S1 File. The initial database search related to the research topic before screening for duplicates and eligibility criteria.**
(DOCX)

**S1 Table. Characteristics of included and excluded articles after removing duplication.**
(DOCX)

**S2 Table. Characteristics of included SRMA studies.**
(XLSX)

**S3 Table. Quality assessment of included SRMA studies using AMSTAR-2 checklist.**
(DOCX)

**S4 Table. Mapped primary studies included in SRMA studies.**
(XLSX)

## Acknowledgments

The authors acknowledge all the authors of the systematic review and meta-analysis included in this umbrella review.
PROSPERO protocol registration
CRD42024511868.

## Author contributions

**Conceptualization:** Neway Ejigu, Negussie Sarbecha, Degefa Gomora, Girma Geta, Chala Kene, Shelema Mengistu, Derese Eshetu, Yaregal Admasu, Telila Mesfin, Daniel Atlaw, Girma Beressa.

**Data curation:** Neway Ejigu, Degefa Gomora, Derese Eshetu, Yaregal Admasu, Telila Mesfin, Daniel Atlaw, Girma Beressa.

**Formal analysis:** Neway Ejigu, Negussie Sarbecha, Telila Mesfin.

**Funding acquisition:** Neway Ejigu.

**Investigation:** Neway Ejigu, Negussie Sarbecha, Chala Kene, Telila Mesfin.

**Methodology:** Neway Ejigu, Negussie Sarbecha, Kenbon Seyoum, Girma Geta, Chala Kene, Shelema Mengistu, Derese Eshetu, Yaregal Admasu, Telila Mesfin, Girma Beressa.

**Project administration:** Neway Ejigu.

**Resources:** Neway Ejigu, Negussie Sarbecha, Kenbon Seyoum, Degefa Gomora, Yaregal Admasu, Girma Beressa.

**Software:** Neway Ejigu, Negussie Sarbecha, Kenbon Seyoum, Degefa Gomora, Chala Kene, Shelema Mengistu, Telila Mesfin, Daniel Atlaw, Girma Beressa.

**Supervision:** Neway Ejigu, Negussie Sarbecha, Kenbon Seyoum, Degefa Gomora, Girma Geta, Chala Kene, Shelema Mengistu, Derese Eshetu, Daniel Atlaw.

**Validation:** Neway Ejigu, Negussie Sarbecha, Degefa Gomora.

**Visualization:** Neway Ejigu, Kenbon Seyoum, Girma Geta, Shelema Mengistu.

**Writing – original draft:** Neway Ejigu.

**Writing – review & editing:** Neway Ejigu, Negussie Sarbecha, Degefa Gomora, Girma Geta, Shelema Mengistu, Derese Eshetu, Daniel Atlaw.

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
