## [Decision Letter · Decision Letter 0]

23 Jul 2024

PGPH-D-24-01423

Prevalence and associated factors of low birth weight in Ethiopia: an umbrella review of a systematic review and meta-analysis

Dear Ejigu,

Thank you for submitting your manuscript to PLOS Global Public Health. After careful consideration, we feel that it has merit but does not fully meet PLOS Global Public Health’s publication criteria as it currently stands. Therefore, we invite you to submit a revised version of the manuscript that addresses the points raised during the review process.

We look forward to receiving your revised manuscript.

Kind regards,

Collins Otieno Asweto, PhD

Academic Editor

Journal Requirements:

1. We noticed you have some minor occurrence of overlapping text with the following previous publication(s), which needs to be addressed:

- https://journals.lww.com/annals-of-medicine-and-surgery/fulltext/2024/02000/neonatal_sepsis_and_its_predictors_in_ethiopia_.52.aspx

- https://www.researchsquare.com/article/rs-1019391/v1

In your revision ensure you cite all your sources (including your own works), and quote or rephrase any duplicated text outside the methods section. Further consideration is dependent on these concerns being addressed.

2. We note that your Data Availability Statement is currently as follows: All relevant data are within the article and its supporting information files.

**Comments to the Author**

1. Does this manuscript meet PLOS Global Public Health’s publication criteria ? Is the manuscript technically sound, and do the data support the conclusions? The manuscript must describe methodologically and ethically rigorous research with conclusions that are appropriately drawn based on the data presented.

Reviewer #1: Partly

Reviewer #2: Yes

2. Has the statistical analysis been performed appropriately and rigorously?

Reviewer #1: Yes

Reviewer #2: No

3. Have the authors made all data underlying the findings in their manuscript fully available (please refer to the Data Availability Statement at the start of the manuscript PDF file)?

Reviewer #1: No

Reviewer #2: No

4. Is the manuscript presented in an intelligible fashion and written in standard English?

Reviewer #1: No

Reviewer #2: No

5. Review Comments to the Author

Reviewer #1: Thank you for giving me the opportunity to review this paper. Low birth weight is an important public health challenge and it is important to coalesce all the data collected on it to help with guiding policy.

In general, the paper will benefit from a round of proof-reading/editing to correct grammatical errors.

Methods

1. Eligibility criteria:

- Line 99-100: It will be good if the authors offered an explanation as to why there were no restrictions to publication dates in their search strategy.

2. Data extraction:

- Did the standardized data extraction form have a section summarizing the author’s overall conclusions? And if not, why?

- How many studies would have been required to have carried out an assessment of publication bias?

Results

1. Characteristics of included systematic review and meta-analysis studies:

- The authors present participants with primary outcome of interest on line 170 then present participants with outcomes of interest. It would be clearer if the authors specify which outcomes of interest they are presenting. The same goes for the number of studies with primary outcome and number of studies with outcomes of interest.

- Line 179: Authors to consider placing the references at the end of the sentence.

- Even though there were no date restrictions for the search strategy, it will be important for the authors to state the time span that their chosen SRMAs covered.

- What is missing from this section is a discussion on the results of the quality assessments of the SMRAs chosen using the AMSTAR-2 tool.

2. Primary studies

- The explanations were not very clear here. For example, when discussing the SRMAs that overlapped, did you mean to say that the 6 SRMAs that included one primary study had 1 primary study in common to all of them? It will also be important to include the reference to this common primary study(ies). Furthermore, when discussing the SRMAs that had no overlap, even though you have referenced them by author name, please add their respective reference numbers.

- Line 196-205: The authors may want to consider breaking down this statement into shorter sentences as it’s quite long in it’s current state and not easily understood.

3. Meta-analysis of predictors of LBW

- The authors may want to consider adding a subtitle to each of the paragraphs with the predictor of interest being discussed.

- Again, please add reference numbers to the papers you are citing.

- Line 250-251: What was the pregnancy interval considered by Habtegiorgis et al?

- Line 255: Please define IFAS as this is the first time the authors are using it in the main paper.

- Line 264-265: What was the maternal age considered by Habtegiorgis et al?

Discussion

- Line 286-291: There is a correlation between LBW and prematurity. It is easier to make the argument of the LBW due to prematurity because the neonate is expected to have a weight lower than 2500g when they are not at term. But no so much for the maturity of the organs.

- Line 299: “Therefore, particular emphasis shall be given to scaling up regular antenatal care

follow up, health education, early detection, and intervention of obstetric complications with the help of Community Health Workers/ Volunteers (CHW/Vs).”. Were the authors intending to give a recommendation here (and not so much what they will be doing following the release of the paper)? If the former, I propose they change ‘shall’ to ‘should’.

- Line 298: The paper being cited for LMICs (Khanal et al) did not look at LMICS in general but specifically studied Nepal.

- Line 317-320: The authors highlight that short pregnancy intervals can result in LBW. But would inadequate maternal intake or depletion of iron and folate alone have a direct significant impact on maternal weight?

Limitations

- This section would benefit from a more nuanced discussion on what types of heterogenicity the authors observed that could be a limitation to this study. And it will also be important to discuss the limitations in terms of the quality of the SRMAs used for this study.

Supplementary Material

- I was unable to open any of the supplementary materials (none of the links worked).

Reviewer #2: Thank you for the opportunity to review this important manuscript. Understanding the prevalence of LBW in Ethiopia is important for informing efforts to address it. The authors have done a very thorough review of the literature, and the manuscript provides an in-depth discussion of factors that put newborns at higher risk of LBW.

I have several suggestions for strengthening the manuscript:

1) My main concern is about the overlap in the primary studies included in the systematic reviews. If I understand correctly, the sum of the number of primary studies across systematic reviews was 259, but only 79 of these were unique, indicating a very high level of overlap. While the authors acknowledge this issue, they still use statistical methods that treat each of the 259 studies as unique. This results in a false level of precision, and also biases the findings towards studies that appear more times in systematic reviews. To make these findings interpretable, I think it will be important to conduct an analysis that removes duplicates.

My other comments are minor:

2) In the introduction, the authors quote a statistic that the global prevalence of LBW is 15.5%, but they also state that the prevalence is 9.8% in sub-Saharan Africa. These numbers come very different sources, so it is hard to compare them. I would recommend using one consistent source for all estimates, such as Okwaraji et al 2024 in the Lancet (National, regional, and global estimates of low birthweight in 2020, with trends from 2000: a systematic analysis) - that study estimates 16.6% global prevalence and 15.7% prevalence in sub-Saharan Africa.

3) In the discussion section, the authors talk about the reasons for the association between prematurity and low birth weight. One reason that is not mentioned is that premature babies have less time in utero and therefore less time to gain weight.

4) In the discussion section, the authors also talk about the relationship between IFA and low birthweight. I would recommend adding a point that this association may be be partly driven by unmeasured confounding, e.g. by socioeconomic status.

5) There are some minor grammatical/language errors throughout the manuscript (e.g., the phrase "limited convulsive information" in the introduction) - a quick review by a language editor (or even just ChatGPT) could help address these.

6. PLOS authors have the option to publish the peer review history of their article (what does this mean? ). If published, this will include your full peer review and any attached files.

**Do you want your identity to be public for this peer review?** For information about this choice, including consent withdrawal, please see our Privacy Policy .

Reviewer #1: **Yes: ** Grace Mambula

Reviewer #2: No

---

## [Decision Letter · Decision Letter 1]

11 Oct 2024

PGPH-D-24-01423R1

Prevalence and associated factors of low birth weight in Ethiopia: an umbrella review of a systematic review and meta-analysis

Dear Neway,

Thank you for submitting your manuscript to PLOS Global Public Health. After careful consideration, we feel that it has merit but does not fully meet PLOS Global Public Health’s publication criteria as it currently stands. Therefore, we invite you to submit a revised version of the manuscript that addresses the points raised during the review process.

We look forward to receiving your revised manuscript.

Kind regards,

Collins Otieno Asweto, PhD

Academic Editor

Journal Requirements:

1. We noticed you have some minor occurrence of overlapping text with the following previous publication(s), which needs to be addressed:

- https://journals.lww.com/annals-of-medicine-and-surgery/fulltext/2024/02000/neonatal_sepsis_and_its_predictors_in_ethiopia_.52.aspx

- https://www.researchsquare.com/article/rs-1019391/v1

In your revision ensure you cite all your sources (including your own works), and quote or rephrase any duplicated text outside the methods section. Further consideration is dependent on these concerns being addressed.

Reviewers' comments:

Reviewer's Responses to Questions

**Comments to the Author**

1. If the authors have adequately addressed your comments raised in a previous round of review and you feel that this manuscript is now acceptable for publication, you may indicate that here to bypass the “Comments to the Author” section, enter your conflict of interest statement in the “Confidential to Editor” section, and submit your "Accept" recommendation.

Reviewer #1: All comments have been addressed

Reviewer #2: (No Response)

2. Does this manuscript meet PLOS Global Public Health’s publication criteria ? Is the manuscript technically sound, and do the data support the conclusions? The manuscript must describe methodologically and ethically rigorous research with conclusions that are appropriately drawn based on the data presented.

Reviewer #1: Yes

Reviewer #2: No

3. Has the statistical analysis been performed appropriately and rigorously?

Reviewer #1: Yes

Reviewer #2: I don't know

4. Have the authors made all data underlying the findings in their manuscript fully available (please refer to the Data Availability Statement at the start of the manuscript PDF file)?

Reviewer #1: Yes

Reviewer #2: No

5. Is the manuscript presented in an intelligible fashion and written in standard English?

Reviewer #1: Yes

Reviewer #2: Yes

6. Review Comments to the Author

Reviewer #1: (No Response)

Reviewer #2: Thank you for the chance to review this revised manuscript.

I have several questions on this version:

1) Missing data analysis - Can you please provide more detailed information on the sensitivity analyses conducted for missing data?

2) Overlap - In the analysis of overlap, you currently use r=134. However, in the text of the manuscript, you say that r=79 (there are 79 unique primary publications). Which is correct? If you do the calculation for CCA using r=79, the result is 18.6, which indicates a vert high degree of overlap. Please clarify this in the next version.

7. PLOS authors have the option to publish the peer review history of their article (what does this mean? ). If published, this will include your full peer review and any attached files.

**Do you want your identity to be public for this peer review?** For information about this choice, including consent withdrawal, please see our Privacy Policy .

Reviewer #1: **Yes: ** Grace Mambula

Reviewer #2: No

---

## [Decision Letter · Decision Letter 2]

28 Nov 2024

PGPH-D-24-01423R2

Prevalence and associated factors of low birth weight in Ethiopia: an umbrella review of a systematic review and meta-analysis

Dear Ejigu,

Thank you for submitting your manuscript to PLOS Global Public Health. After careful consideration, we feel that it has merit but does not fully meet PLOS Global Public Health’s publication criteria as it currently stands. Therefore, we invite you to submit a revised version of the manuscript that addresses the points raised during the review process.

We look forward to receiving your revised manuscript.

Kind regards,

Collins Otieno Asweto, PhD

Academic Editor

Journal Requirements:

1. We noticed you have some minor occurrence of overlapping text with the following previous publication(s), which needs to be addressed:

- https://journals.lww.com/annals-of-medicine-and-surgery/fulltext/2024/02000/neonatal_sepsis_and_its_predictors_in_ethiopia_.52.aspx

- https://www.researchsquare.com/article/rs-1019391/v1

In your revision ensure you cite all your sources (including your own works), and quote or rephrase any duplicated text outside the methods section. Further consideration is dependent on these concerns being addressed.

Reviewers' comments:

Reviewer's Responses to Questions

**Comments to the Author**

1. If the authors have adequately addressed your comments raised in a previous round of review and you feel that this manuscript is now acceptable for publication, you may indicate that here to bypass the “Comments to the Author” section, enter your conflict of interest statement in the “Confidential to Editor” section, and submit your "Accept" recommendation.

Reviewer #3: All comments have been addressed

Reviewer #4: All comments have been addressed

Reviewer #5: (No Response)

Reviewer #6: All comments have been addressed

Reviewer #7: All comments have been addressed

2. Does this manuscript meet PLOS Global Public Health’s publication criteria ? Is the manuscript technically sound, and do the data support the conclusions? The manuscript must describe methodologically and ethically rigorous research with conclusions that are appropriately drawn based on the data presented.

Reviewer #3: Partly

Reviewer #4: Yes

Reviewer #5: Yes

Reviewer #6: Yes

Reviewer #7: Yes

3. Has the statistical analysis been performed appropriately and rigorously?

Reviewer #3: Yes

Reviewer #4: Yes

Reviewer #5: I don't know

Reviewer #6: I don't know

Reviewer #7: Yes

4. Have the authors made all data underlying the findings in their manuscript fully available (please refer to the Data Availability Statement at the start of the manuscript PDF file)?

Reviewer #3: Yes

Reviewer #4: Yes

Reviewer #5: Yes

Reviewer #6: Yes

Reviewer #7: Yes

5. Is the manuscript presented in an intelligible fashion and written in standard English?

Reviewer #3: Yes

Reviewer #4: Yes

Reviewer #5: Yes

Reviewer #6: Yes

Reviewer #7: Yes

6. Review Comments to the Author

Reviewer #3: TOPIC

Prevalence and associated factors of low birth weight in Ethiopia: an umbrella review of a systematic review and meta-analysis

Remark(s) 1: This should read: “Prevalence and associated factors of low birth weight in Ethiopia: an umbrella review of systematic reviews and meta-analyses

ABSTRACT

Background:

Lines 19-21: Despite the reduction of neonatal morbidity and mortality, is a major goal needed to be achieved by 2030, the burden remains a major challenge for low-income countries including Ethiopia.

Remark(s) 2: This does not read well, rephrase.

Lines 21-23: Although various systematic review and meta-analysis (SRMA) studies have been conducted on LBW in Ethiopia, there is notable variation among their findings.

Remark(s) 3: Recast to read: “Although various systematic reviews and meta-analyses (SRMA) have been conducted on LBW in Ethiopia, there is notable variation among their findings.

Aim: Lines 23-25: Therefore, an umbrella review of these studies aimed to pool the inconsistent findings into a single summary estimate that could guide health policy development and planning in Ethiopia.

Remark(s) 4: This is inadequate…a mere summary of inconsistent findings from systematic reviews and meta-analyses does not provide a strong justification for this review. Note that this is a review of systematic reviews and meta-analyses, all of which are evidence synthesis of primary studies conducted. Typically, reviews are conducted to synthesis evidence, with the view to determining the extent, scope, and direction of existing evidence on a particular subject area which may also help to unveil gaps in literature.

KEY WORDS

Remark(s) 5: The “Umbrella review” not a MeSH Term for this topic and should be replaced with a MeSH Term.

INTRODUCTION

Page 1, para 1, lines 48 & 49: LBW is a useful public health measure of poverty, healthcare delivery, nutrition, and maternal health (2).

Remark(s) 6: Recast to read: “LBW is a useful public health measure of poverty, quality of healthcare delivery, quality of nutrition, and quality of maternal health (2).

Page 1, para 1, lines 50-53: LBW infants are more….

Remark(s) 7: Recast to read: “Infants with LBW are more….

Page 1, para 2, lines 57-59: …, low birth weight newborns are about 20 times more likely to die than with normal birth weight (10).

Remark(s) 8: Recast to read: “…, newborns with low birth weight are about 20 times more likely to die than with normal birth weight (10).

Page 2, para 2, line 71: Reduction of neonatal mortality is one of the major Sustainable Developmental Goals (SDGs) in a country like Ethiopia….

Remark(s) 9: Provide details on the specific SDG you are referring to.

Page 2, para 2, lines 74-79: Despite extensive attempts by the Ministry of Health (MoH) and non-governmental organizations (NGOs), LBW is still one of the leading causes of neonatal morbidity and mortality in Ethiopia. This may be due to limited information available about the problem. Hence, the availability of local information on the determinant factors of LBW has a major role in the management and control of the case in the country and gaining insights of such factors from this study will help the stake holders and implementers to design new as well as to strength the existing programs towards the problem.

Remark(s) 10: Several primary studies and systematic reviews and meta-analyses may have provided a reasonably adequate information on the subject, at least regarding the factors that account for LBW in Ethiopia. This cannot be the primary focus of this review. Rephrase this portion.

Remark(s) 11: “Hence, the availability of local information on the determinant factors of LBW has a major role….” Recast this to read: “Therefore, the availability of local information on the determinants of LBW has a major role….”

Page 2, para 2, lines 79-82: Even though, there are many systematic review and meta-analysis in Ethiopia but their finding is inconsistent, therefore, umbrella review of these studies aimed to pool the findings into a single summary estimate that can provide guidance for health policy development and planning in Ethiopia.

Remark(s) 12: Recast to read: “Even though, there are many systematic reviews and meta-analyses in Ethiopia, the findings are inconsistent, therefore, an umbrella review….

Remark(s) 13: The justification for the review is insufficient….there is need to go beyond the supposed inconsistencies in the existing reviews. The authors need to consult literature on why review of reviews are conducted. Note, you are reviewing systematic reviews and meta-analyses, which may have addressed the supposed inconsistencies found in the primary studies used.

METHODS

Page 3, lines 86-89: This method provides decision-makers with a comprehensive source of information to support clinical recommendations and guide policymakers, intervention developers, and healthcare evaluators based on prior research (26,27).

Remark(s) 14: This is irrelevant for the section, delete and provide a more relevant information.

Eligibility criteria:

Remark(s) 15: The authors should respond to the following:

Were all the included reviews “systematic reviews and meta-analyses”? Did you include scoping reviews? Why have the authors included studies that have not passed peer-review in this review?

RESULTS

Remark(s) 16: Provide a table to demonstrate how findings of the current review compare or differ from findings of the included systematic reviews and meta-analyses. A column in the table should show the novelty of the current review.

DISCUSSION

Page 6, para 2, lines 295 & 296: The five SRMA in this umbrella review revealed that the overall pooled prevalence of LBW in Ethiopia was 16% (95% CI 13, 18).

Remark(s) 17: Recast to read: “The five SRMA in this umbrella review (cite the five SRMA here) revealed that the overall pooled prevalence of LBW in Ethiopia was 16% (95% CI 13, 18).

Page 6, para 2, lines 296-298: The finding of this study agreed with studies conducted in Africa (49) and the global systematic analysis estimates (13). However, the result of this study was higher than those of studies conducted in SSA countries (50), and Iran (51).

Remark(s) 18: These sentences do not read well, recast.

Page 6, para 3, lines 301-303: Those infants who delivered before 37 weeks of gestation were 7.86 times more likely to have LBW compared to those delivered after 37 weeks of gestation.

Remark(s) 19: Recast to read: “Infants who were delivered before 37 weeks….”

Page 6, para 3, lines 304 & 305: This might be due to the fact that several organ of the fetus usually become mature by the end of 37 weeks of gestation.

Remark(s) 20: This does not read well, recast.

Page 7, para 1, lines 307-309: The odds of having LBW among women who did not attend ANC follow-up were 2.4 times higher than who did attend ANC.

Remark(s) 21: Introduce the word “those” in between the words “than” and “who”.

Page 7, para 2, line 317: In this study, women with PIH had 4.20 times higher odds of giving LBW compared to normotensive women.

Remark(s) 22: Recast to read: “…higher odds of giving birth to babies with LBW….” Or “…higher odds of delivering babies with LBW….” Apply this throughout the manuscript.

Page 7, para 2, lines 317-321: In this study, women with PIH had 4.20 times higher odds of giving LBW compared to normotensive women. This finding is in line with the WHO secondary analysis survey conducted in LMICs (56), Indonesia (52), and Malaysia (54). This might be due to endothelial cell injury and vasoconstriction of blood vessels, resulting in reduced utero-placental blood perfusion leading to LBW (57).

Remark(s) 23: There is a problem with this paragraph. “This might be due to endothelial cell injury and vasoconstriction of blood….” Are the authors seeking to explain their finding or those of the WHO secondary surveys (52,54,56)? This can leave the audiences confused and must be corrected.

Rather, the point should read: “In this study, women with PIH had 4.20 times higher odds of giving LBW compared to normotensive women. This might be due to endothelial cell injury and vasoconstriction of blood vessels, resulting in reduced utero-placental blood perfusion leading to LBW (57). This finding is in line with the WHO secondary analysis survey conducted in LMICs (56), Indonesia (52), and Malaysia (54). There are similar errors of construction throughout the discussion which must be corrected to guarantee clarity.

Page 8, para 1, lines 331-331: Women who gave birth before 24 months had 2.96 times higher odds of having LBW compared to those with birth interval greater than 24 months.

Remark(s) 24: Recast to read: “…had 2.96 times higher odds of delivering babies with LBW compared….” Correct this throughout the manuscript.

LIMITATION OF THE REVIEW

Remark(s) 25: The section is woefully inadequate; there are more issues that must be reported. For instance, inclusion of unpublished reviews raises concerns over the reliability of the study.

CONCLUSION

Remark(s) 26: The section is inadequate; provide a more comprehensive conclusion to cover all the key findings and also add recommendations. Note that the recommendations must be directed at specific organizations and individuals for action. Additionally, you must suggest possible areas for future research direction in the area.

COMMENTS FOR THE AUTHORS

Overall, the authors have made a good effort at dealing with a subject of high public health importance. However, the paper will need substantial revision to make it suitable for publication. Apart from the remarks provided above, the authors must conduct a thorough and rigorous English Language editing of the paper.

Reviewer #4: Check for Grammatical errors

Reviewer #5: I have attached my comments

Reviewer #6: Dear Editor,

I found that only minor revisions were required in the first round. Having joined as a reviewer for the second round, I can confirm that the previously highlighted section has been adequately revised.

Overall, the manuscript is of good quality and provides valuable information. I believe it is now suitable for publication.

Kind regards,

Ishrat

Reviewer #7: The authors have addressed the concerns raised by other reviewers satisfactorily.

7. PLOS authors have the option to publish the peer review history of their article (what does this mean? ). If published, this will include your full peer review and any attached files.

**Do you want your identity to be public for this peer review?** For information about this choice, including consent withdrawal, please see our Privacy Policy .

Reviewer #3: **Yes: ** BOTHA, Nkosi Nkosi

Reviewer #4: No

Reviewer #5: No

Reviewer #6: No

Reviewer #7: **Yes: ** PRISCILIA UHUANMWEN IMADE

---

## [Decision Letter · Decision Letter 3]

2 Jan 2025

PGPH-D-24-01423R3

Prevalence and associated factors of low birth weight in Ethiopia: an umbrella review of a systematic review and meta-analysis

Dear Ejigu,

Thank you for submitting your manuscript to PLOS Global Public Health. After careful consideration, we feel that it has merit but does not fully meet PLOS Global Public Health’s publication criteria as it currently stands. Therefore, we invite you to submit a revised version of the manuscript that addresses the points raised during the review process.

We look forward to receiving your revised manuscript.

Kind regards,

Collins Otieno Asweto, PhD

Academic Editor

Journal Requirements:

Additional Editor Comments (if provided):

Reviewers' comments:

Reviewer's Responses to Questions

**Comments to the Author**

1. If the authors have adequately addressed your comments raised in a previous round of review and you feel that this manuscript is now acceptable for publication, you may indicate that here to bypass the “Comments to the Author” section, enter your conflict of interest statement in the “Confidential to Editor” section, and submit your "Accept" recommendation.

Reviewer #3: All comments have been addressed

Reviewer #4: All comments have been addressed

Reviewer #5: All comments have been addressed

2. Does this manuscript meet PLOS Global Public Health’s publication criteria ? Is the manuscript technically sound, and do the data support the conclusions? The manuscript must describe methodologically and ethically rigorous research with conclusions that are appropriately drawn based on the data presented.

Reviewer #3: Partly

Reviewer #4: Yes

Reviewer #5: Yes

3. Has the statistical analysis been performed appropriately and rigorously?

Reviewer #3: Yes

Reviewer #4: Yes

Reviewer #5: Yes

4. Have the authors made all data underlying the findings in their manuscript fully available (please refer to the Data Availability Statement at the start of the manuscript PDF file)?

Reviewer #3: Yes

Reviewer #4: Yes

Reviewer #5: No

5. Is the manuscript presented in an intelligible fashion and written in standard English?

Reviewer #3: Yes

Reviewer #4: Yes

Reviewer #5: No

6. Review Comments to the Author

Reviewer #3: The authors have done well in repsonding to the comments but not as expected. The justification for the review is weak and needs further revision, the table on Remark(s) 16: Provide a table to demonstrate how findings of the current review compare or differ from findings of the included systematic reviews and meta-analyses. A column in the table should show the novelty of the current review, was not adequately done. In fact, the table provided by the authors is not clear and must be revised with further details.

Reviewer #4: Nil

Reviewer #5: I have attached my comments within.

7. PLOS authors have the option to publish the peer review history of their article (what does this mean? ). If published, this will include your full peer review and any attached files.

**Do you want your identity to be public for this peer review?** For information about this choice, including consent withdrawal, please see our Privacy Policy .

Reviewer #3: **Yes: ** BOTHA, Nkosi Nkosi

Reviewer #4: No

Reviewer #5: No

---

## [Decision Letter · Decision Letter 4]

7 Apr 2025

Prevalence and associated factors of low birth weight in Ethiopia: an umbrella review of a systematic review and meta-analysis

PGPH-D-24-01423R4

Dear Mr. Ejigu,

We are pleased to inform you that your manuscript 'Prevalence and associated factors of low birth weight in Ethiopia: an umbrella review of a systematic review and meta-analysis' has been provisionally accepted for publication in PLOS Global Public Health.

Best regards,

Julia Robinson

Executive Editor

Reviewer Comments (if any, and for reference):

Reviewer's Responses to Questions

**Comments to the Author**

1. If the authors have adequately addressed your comments raised in a previous round of review and you feel that this manuscript is now acceptable for publication, you may indicate that here to bypass the “Comments to the Author” section, enter your conflict of interest statement in the “Confidential to Editor” section, and submit your "Accept" recommendation.

Reviewer #1: All comments have been addressed

2. Does this manuscript meet PLOS Global Public Health’s publication criteria ? Is the manuscript technically sound, and do the data support the conclusions? The manuscript must describe methodologically and ethically rigorous research with conclusions that are appropriately drawn based on the data presented.

Reviewer #1: Yes

3. Has the statistical analysis been performed appropriately and rigorously?

Reviewer #1: Yes

4. Have the authors made all data underlying the findings in their manuscript fully available (please refer to the Data Availability Statement at the start of the manuscript PDF file)?

Reviewer #1: Yes

5. Is the manuscript presented in an intelligible fashion and written in standard English?

Reviewer #1: Yes

6. Review Comments to the Author

Reviewer #1: (No Response)

7. PLOS authors have the option to publish the peer review history of their article (what does this mean? ). If published, this will include your full peer review and any attached files.

**Do you want your identity to be public for this peer review?** For information about this choice, including consent withdrawal, please see our Privacy Policy .

Reviewer #1: **Yes: ** Grace Mambula
